# PROCEEDINGS A

## Invited reply

mechanics

material surfaces, rectifying developables, Möbius bands

**Author for correspondence:**
Eliot Fried
e-mail: Eliot.fried@oist.jp

# Reply to the comment of van der Heijden and Starostin

Yi-Chao Chen[1], Roger Fosdick[2] and Eliot Fried[3]

[1]Department of Mechanical Engineering, University of Houston, Houston, TX 77204-4006, USA
[2]Department of Aerospace Engineering and Mechanics, University of Minnesota, Minneapolis, MN 55455-0153, USA
[3]Mechanics and Materials Unit, Okinawa Institute of Science and Technology Graduate University, Okinawa, Japan 904-0495

Y-CC, 0000-0002-7236-5581; EF, 0000-0001-5329-3394

## 1. Technical overview in response to the comments of van der Heijden & Starostin [1]

The comment of van der Heijden & Starostin [1] concerning the contents of the papers of Chen & Fried [2] and of Chen *et al.* [3–5] and their relationship to the paper of Starostin & van der Heijden [6] contains information that was not included in Starostin & van der Heijden [6]. That new information clarifies that a rectifying developable parametrization of a reference strip is their concern and that they introduce supplemental conditions to ensure that the strip is a rectangle. These augmentations are welcome and we shall refer to them in the following.

Our efforts in publications [2–5] have been to understand previously unexplained issues surrounding the relevance of rectifying developable parametrizations to problems involving deformations of flat unstretchable material surfaces. In the following, we briefly address the major issues which we believe have stimulated the authors of [1], and we give reasons for why the new information provided therein does not resolve the salient concerns. We maintain that the variational problem of determining an energy minimizing deformation of an unstretchable material rectangle to a free-standing Möbius band with a 180° twist remains unsolved.

The major issues with the strategy of Starostin & van der Heidjen [6], reprised in their comment [1], stem from their reliance on the class of rectifying developable parameterizations of a rectangular region

The accompanying comment can be viewed at https://dx.doi.org/10.1098/rspa.2021.0629.

without introducing and properly defining:

  (i) a single, fixed material reference configuration from which to measure the deformation of a two-dimensional elastic body;

  (ii) a suitable class of competitors and related variations when formulating a solid mechanics problem within the calculus of variations.

We sketch the basis for our line of argument here, relegating details and further clarifying discussion to subsequent §3a–3c. In §2, we briefly appraise the claims of van der Heijden & Starostin [1] concerning the papers of Chen and Fried [2] and Chen *et al.* [3–5].

Equation (2) from the comment [1], which does not appear anywhere in their paper [6], involves the introduction of a fixed reference configuration for a material rectangle of length $L$ and width $2w$ by labelling each material point $X$ using the pair $(u, v)$ of rectangular Cartesian coordinates: $X = u e_3 + v e_1$, $(u, v) \in [0, L] \times [-w, w]$. Prior to this, they extract, from [6], their equation (1) to serve as a representative surface, say $\mathcal{S}$, in their competing class of rectifying developable Möbius bands parameterized by the pair $(s, v) \in [0, L] \times [-w, w]$. They then set out to interpret any placement $\mathcal{S}$ of the form defined in their equation (1) as a deformation from a supposedly unique planar material reference configuration. As a first step, they recognize in [1] that the form $s e_3 + v(e_1 + \eta(s)e_3)$, $s \in [0, L]$, $v \in [-w, w]$ covers a fixed rectangular shape of length $L$ and width $2w$, corresponding to the planar development of $\mathcal{S}$ if $\eta$ satisfies $\eta(0) = 0$ and $\eta(L) = 0$, two equations which also do not appear anywhere in [6]. Then, in the second part of their equation (2) they link the fixed pair $(u, v)$ to the pair $(s, v)$ by invoking the relation $u e_3 + v e_1 = s e_3 + v(e_1 + \eta(s)e_3)$. But this requires that the material point $X = u e_3 + v e_1$, defined in terms of the supposedly *fixed* pair $(u, v)$, must, in contradiction, depend upon $\eta(s)$, a function that changes with each competitor Möbius band $\mathcal{S}$ of their form (1). Although the planar development with rectangular shape of length $L$ and width $2w$ corresponding to any placement $\mathcal{S}$ of the form defined in their (1) is fixed, the location of material points $X = u e_3 + v e_1$ within that rectangle depends upon $\mathcal{S}$ through the presence of the function $\eta$. Thus, the strategy of van der Heijden & Starostin outlined in [1] and followed in Starostin & van der Heijden [6] is not based upon the notion of a fixed material reference configuration.

In formulating their variational problem, Starostin & van der Heijden [6] thus overlooked the fundamental requirement that the deformation of each competitive placement of the form of their (1) must be measured from the *same* fixed rectangular reference configuration, say $\mathcal{R}$, with material points $X = u e_3 + v e_1$, $u \in [0, L]$, $v \in [-w, w]$, which defines the rectangular material strip. Instead, as explained in [1], they connected each pair $(u, v)$ to the pair $(s, v)$ as indicated in the second part of their (2). Consequently, in [6] Starostin and van der Heijden measured the deformation of each competitive placement $\mathcal{S}$ of the form of their (1) relative to a reference configuration $\mathcal{R}'$ of the same rectangular shape as $\mathcal{R}$ but whose points $X' := s e_3 + v(e_1 + \eta(s)e_3)$, $s \in [0, L]$, $v \in [-w, w]$, depend upon and, because of the presence of the function $\eta$, change with each particular placement $\mathcal{S}$ that they considered. Because of this, it follows that the rectifying developable mapping of each reference configuration $\mathcal{R}'$ to the associated placement surface $\mathcal{S}$ of the form (1) chosen by Starostin & van der Heijden [6] is an isometric mapping, as van der Heijden & Starostin [1] emphatically note in their comment. But the detailed calculation in that comment only confirms that the fixed rectangular reference configuration $\mathcal{R}$ and each particular placement $\mathcal{S}$ share a mutual isometry relation, namely that $\mathcal{R}$ is the planar development of any placement $\mathcal{S}$ of the form defined in their equation (1), not that every $\mathcal{S}$ defined in (1) is the isometric deformation of the same fixed material reference configuration $\mathcal{R}$.

In their variational problem, Starostin & van der Heidjen [6] envision a unstretchable fixed rectangular material strip of length $L$ and width $2w$, and consider 180° twist Möbius bands of the rectifying developable form $\mathcal{S}$ defined in [1], to be the admissible class of competing curved

surfaces in the computation of their energy functional. However, to formulate a variational strategy for analysing the minimal energy configurations of a solid material surface, the fixed material coordinates $(u, v)$ of $X = ue_3 + ve_1$, $u \in [0, L]$, $v \in [-w, w]$, that define the fixed material reference configuration $\mathcal{R}$ relative to which a deformation is to be computed, needs to be related to the parameters $(s, v)$, $s \in [0, L]$, $v \in [-w, w]$, that are used to define every admissible placement $\mathcal{S}$. While the approach to do this promoted by van der Heijden & Starostin [1], and described above, leads to a contradiction, an acceptable alternative, natural in the field of variational calculus, would be to assume that some surface $\bar{\mathcal{S}}$ to be determined, defined by

$$\bar{x} = \bar{x}(s, v) = \bar{r}(s) + v[\bar{b}(s) + \bar{\eta}(s)\bar{t}(s)], \quad s \in [0, L], \ v \in [-w, w], \tag{A}$$

which is of the form of (1) in van der Heijden & Starostin [1], is the minimizer that they seek, and, noting that the function $\bar{\eta}$ is fixed, to introduce

$$X = ue_3 + ve_1 = se_3 + v(e_1 + \bar{\eta}(s)e_3), \tag{B}$$

from which the parameter $s$ that appears in every competing placement of the form (1) chosen as admissible in [6] is to be determined in terms of the pair $(u, v)$. The consequence of introducing (B) is that, aside from the minimizer (A), all other competitor placements $\mathcal{S}$ of the form (1) of [6] will suffer a surfacial shear deformation relative to the fixed reference configuration $\mathcal{R}$ and, thus, are not isometric deformations, as the constraint of unstretchability requires. Effectively, the variational strategy of Starostin & van der Heidjen [6] and further discussed in van der Heijden & Starostin [1] involves a reference configuration $\mathcal{R}'$ that, as noted above, changes with each admissible 180° twist Möbius band $\mathcal{S}$ of the form (1) of [6]. However, comparing the energies of two configurations of a material surface is futile unless those energies are measured relative to the same chosen material reference configuration.

An objective justification of the above comments is provided in §3. In the next §2, we briefly address the criticisms of our publications [2–5] that van der Heijden & Starostin [1] express.

## 2. Response to the criticisms of publications [2–5]

Chen & Fried [2] do not explicitly state that the solution of Starostin & van der Heijden [6] is incorrect. Instead, they explore the possibility of whether the class of rectifying developable parametrizations of an unstretchable rectangular material strip is sufficiently broad to describe all configurations obtained by isometrically deforming such a strip. All conclusions presented in §7 of the paper of Chen & Fried [2] follow without error from the hypotheses set forth in §4 of that same paper. This includes the 'main result' singled out by van der Heijden & Starostin [1]. The hypotheses of Chen & Fried [2] involve treating a rectifying developable parametrization as a deformation of an unstretchable rectangular material strip. That choice was made without the newly provided condition (2) of van der Heijden & Starostin [1], which merely introduces a single rectangular *planar development* connected to their complete class of rectifying developable parameterizations.

Chen *et al.* [3] focus on developing a two-dimensional theory for equilibrated deformations of unstretchable material surfaces. Rectifying developable parametrizations are not mentioned. The formulation or solution strategy of Starostin & van der Heijden [6] are also not mentioned. Starostin & van der Heijden [6] are cited, along with other authors, only to establish precedents for specializing the results of Chen *et al.* [3] to the situation in which the elastic energy density of a material surface is proportional to the square of the mean curvature of the deformed surface. There is no basis for the claim that Chen *et al.* [3] elaborate upon the work of Chen & Fried [2]. The observations of van der Heijden & Starostin [1] have no bearing on the work of Chen *et al.* [3].

Chen *et al.* [4] obtain a representation for the isometric deformation of a rectangular material surface. In their §5, which concerns alternative representations, and the first paragraph of their final section, they mention issues that relate to rectifying developable parameterizations and the

need to identify a fixed reference configuration. Those concerns are closely related to the issues summarized in §1 and discussed below with justifying context in §3a–3c.

Chen *et al.* [5] focus on matters related to maintaining the identity of material points when a material surface is subjected to an isometric deformation. These matters are addressed completely independent of the formulation or solution strategy of Starostin & van der Heijden [6], a work that is, accordingly, not cited. This issue does, however, arise in connection with rectifying developable parameterizations, as Chen *et al.* [4] demonstrate, and we return to this issue in the final paragraph of §3c.

The comments contained in van der Heijden & Starostin [1] do not point to any specific errors in the papers of Chen & Fried [2] and Chen *et al.* [3–5]. By introducing an undistorted fixed, rectangular reference configuration through the first equation of their newly provided condition (2), van der Heijden & Starostin [1] do expose a promising alternate strategy. However, this was not stated prior to the work of Chen & Fried [2] and does not invalidate the self-consistent results of their work, which clearly states and carefully adheres to their expressed hypotheses. Moreover, in applying the second part of their (2), and using their newly stated requirement $\eta(0) = \eta(L) = 0$, van der Heijden & Starostin [1] introduce a rectangular reference configuration of length $L$ and width $2w$ for the identification of the material points that changes with each rectifying developable parameterization covered by their (1). Using their strategy, the *shape* of the rectangle is fixed but not the position of its internal material points, relative to which the deformation of a placement (1) is to be measured.

## 3. Detailed response concerning the work of Starostin & van der Heijden [6]

Here, we present the basis upon which we have earlier discussed the work of Starostin & van der Heijden [6]. In §3a, we briefly review the background material that is not contentious, but which is essential for the purpose of clarity. Using this, in §3b, we argue our point of contention with the strategy of Starostin & van der Heijden [6], including the clarification presented in the comments of van der Heijden & Starostin [1]. We end, in §3c, with a summarizing discussion.

### (a) Background on rectifying developable parametrizations and the Wunderlich functional

The fundamental problem of determining the equilibrium configuration of a free-standing Möbius band that is formed from the deformation of a flat unstretchable rectangular material strip by joining its two ends together after a 180° twist has a long history dating back to Sadowsky [7,8] and Wunderlich [9,10], and many papers have been written about this issue. However, to date it remains unsolved.

Most recently, Starostin & van der Heijden [6],[1] with the aim of solving this problem, considered the variational problem of minimizing the Wunderlich [9,10] functional over the set of smooth rectifying developable Möbius bands, of fixed length $L$ and width $2w$, each member $\mathcal{S}$ of which has a parametrization

$$\left.\begin{array}{l} \boldsymbol{y} = \boldsymbol{y}(s,t) = \boldsymbol{r}(s) + t[\boldsymbol{b}(s) + \eta(s)\boldsymbol{t}(s)] \in \mathbb{E}^3 \\ \tau(s) = \eta(s)\kappa(s), \quad s \in [0,L], \ t \in [-w,w]. \end{array}\right\} \tag{3.1}$$

and

In (3.1), $\boldsymbol{r}$ is the midline directrix, $\boldsymbol{t}$, $\boldsymbol{b}$, $\kappa$ and $\tau$ are the corresponding unit tangent, unit binormal, curvature and torsion, and $\mathbb{E}^3$ is three-dimensional point space. Of course, $\boldsymbol{t} := \boldsymbol{r}'$, $\boldsymbol{b} := \boldsymbol{t} \times \boldsymbol{p}$, $\kappa := |\boldsymbol{t}'|$ and $\tau := \boldsymbol{t} \cdot (\boldsymbol{p} \times \boldsymbol{p}')$, where the normal to the directrix is given by $\boldsymbol{p} := \boldsymbol{t}'/\kappa$, and the Frenet–Serret equations hold: $\boldsymbol{t}' = \kappa\boldsymbol{p}, \boldsymbol{p}' = -\kappa\boldsymbol{t} + \tau\boldsymbol{b}, \boldsymbol{b}' = -\tau\boldsymbol{p}$. The shape of each Möbius band $\mathcal{S}$ is thus completely determined by its midline directrix, which is smooth and satisfies $\boldsymbol{r}(0) = \boldsymbol{r}(L)$ with smooth connection at its closure point. To ensure closure with a 180° twist, $\boldsymbol{t}(L) = \boldsymbol{t}(0)$ and $\boldsymbol{b}(L) = -\boldsymbol{b}(0)$. The generatrix $\boldsymbol{g}$ of $\mathcal{S}$ intersects the midline directrix $\boldsymbol{r}$ at each $s \in [0,L]$ and is

---

[1]In [1], van der Heijden and Starostin augmented and clarified the basic strategy that they applied in [6]. Thus, we consider the amendments to be an integral part of their work in [6].

given by $g(s,t) := t[b(s) + \eta(s)t(s)]$ for $t \in [-w, w]$. For $\mathcal{S}$ to be smooth, it is thus necessary that $g(L,t) = -g(0,t)$ for $t \in [-w, w]$ and, consequently, that $\eta(L) = -\eta(0)$.

Being developable, $\mathcal{S}$ can be severed along the generator $g(0,t)$ through its closure point and isometrically rolled out onto a plane. This shows that the corresponding subset of the plane and $\mathcal{S}$ satisfy a mutual isometry relation. Thus, there is an isometric mapping of $\mathcal{S}$ into a planar strip $\mathcal{D}$ of midline length $L$ and width $2w$ and it is given by the parameterization

$$
\left.
\begin{aligned}
x = x(s,t) = (s + t\eta(s))\imath_1 + t\imath_2 \in \mathbb{E}^3 \\
\tau(s) = \eta(s)\kappa(s), \quad s \in [0, L], \ t \in [-w, w],
\end{aligned}
\right\} \tag{3.2}
$$

and

where $\{\imath_1, \imath_2\}$ denotes an orthonormal basis in the plane of $\mathcal{D}$.[2] The isometric mapping $x \in \mathcal{D} \mapsto y \in \mathcal{S} \subset \mathbb{E}^3$ of the points is defined implicitly via the parameter pair $(s, t)$, together with the assignment that the single closure point $r(0) = r(L)$ of the midline directrix in $\mathcal{S}$ corresponds to the two points $x(0,0) := o$ and $x(L,0) = L\imath_1$ in $\mathcal{D}$. The ends of the strip $\mathcal{D}$ in (3.2) are thus given, respectively, by $t\eta(0)\imath_1 + t\imath_2$ and $(L + t\eta(L))\imath_1 + t\imath_2$ for $t \in [-w, w]$, where $\eta(L) = -\eta(0)$, which has the consequence that $\mathcal{D}$ has the shape of an isosceles trapezoid, being rectangular only if $\eta(L) = -\eta(0) = 0$, which, for definiteness, we shall assume hereinafter.

The Wunderlich [9,10] functional is based on the reduction of an area integral that characterizes, in a mechanics setting, the bending energy of an unstretchable material strip in terms of a line integral. For a developable Möbius band $\mathcal{S}$, the bending energy is proportional to

$$
\int_{\mathcal{S}} H^2 \, da, \tag{3.3}
$$

where $H$ and $da$ denote the mean curvature and element of surface area on $\mathcal{S}$. If, in addition, $\mathcal{S}$ has a rectifying developable parametrization (3.1), then $H = \kappa(1 + \eta^2(s))/2(1 + \eta'(s)t)$, $da = (1 + \eta'(s)t) \, ds \, dt$, and the area integral in (3.3) is thus converted to an iterated integral over the parameter space, followed by explicit integration over the interval $t \in [-w, w]$ across the generator at each fixed parameter value $s$. This results in the Wunderlich functional

$$
\mathcal{W} = \frac{w}{4} \int_0^L \frac{\kappa^2(1 + \eta^2)^2}{w\eta'} \log\left(\frac{1 + w\eta'}{1 - w\eta'}\right) ds. \tag{3.4}
$$

To evaluate $\mathcal{W}$, for any rectifying developable Möbius band (3.1), a midline directrix $r$ is needed—one that is smooth and satisfies $r(0) = r(L)$ with smooth connection at its closure point, so that $t(L) = t(0)$ and $b(L) = -b(0)$ to accommodate closure with a 180° twist, and corresponds to $\eta(L) = -\eta(0) = 0$. Then, as noted earlier, the fields $\kappa$ and $\eta$ may be calculated and the value of $\mathcal{W}$ is determined.

## (b) Set-up and critique of the variational approach based on rectifying developable parameterizations

Starostin & van der Heijden [6] apply a variational scheme to the functional $\mathcal{W}$ defined in (3.4), and reportedly determine, within the (smooth) set of rectifying developable 180° twist Möbius bands as given in (3.1) with $\eta(L) = -\eta(0) = 0$ and described thereafter, the equilibrium shape $\mathcal{S}_*$,

$$
\left.
\begin{aligned}
y_* = y_*(s,t) = r_*(s) + t[b_*(s) + \eta_*(s)t_*(s)] \in \mathbb{E}^3 \\
\tau_*(s) = \eta_*(s)\kappa_*(s), \quad s \in [0, L], \ t \in [-w, w],
\end{aligned}
\right\} \tag{3.5}
$$

and

of a free-standing *unstretchable* Möbius band in the absence of tractions or couple tractions on its edge. Analogous to the various descriptors of $\mathcal{S}$, the descriptors $r_*$, $t_*$, $b_*$, $\kappa_*$ and $\tau_*$ of $\mathcal{S}_*$ must satisfy $t_* := r'_*$, $b_* := t_* \times p_*$, $\kappa_* := |t'_*|$, and $\tau_* := t_* \cdot (p_* \times p'_*)$, where the unit normal to the directrix is given by $p_* := t'_*/\kappa_*$, and, of course, the Frenet–Serret equations hold and $\eta_*(L) = -\eta_*(0) = 0$.

---

[2]The mapping $(s,t) \mapsto x$ in (3.2) is injective provided $1 + t\eta'(s) > 0$ for $s \in [0, L]$ and $t \in [-w, w]$.

It is essential that the closure points for all $\mathcal{S}$ must be the same, and satisfy $r(0) = r(L) = r_*(0) = r_*(L)$, and, naturally, for smoothness with a 180° twist, that the closure conditions for all $\mathcal{S}$ are assigned such that $t(L) = t(0) = t_*(L) = t_*(0)$, $b(0) = -b(0) = b_*(L) = -b_*(0)$. In addition, the generators through the closure points for all $\mathcal{S}$ are necessarily restricted such that $\eta(L) = -\eta(0) = \eta_*(L) = -\eta_*(0) = 0$. Thus, the ostensible optimal $\mathcal{S}_*$ and any competitor $\mathcal{S}$ must share the same closure point at $s = 0$ and $s = L$, and the corresponding generators through that closure point must coincide with the same straight line segment of length $2w$ in $\mathbb{E}^3$.

As with $\mathcal{S}$, it follows that $\mathcal{S}_*$ satisfies a mutual isometry relation with a flat region, meaning that there is an isometric mapping of $\mathcal{S}_*$ into a rectangular strip $\mathcal{D}_*$ of length $L$ and width $2w$ with parameterization

$$x_* = x_*(s,t) = (s + t\eta_*(s))\imath_1 + t\imath_2 \in \mathbb{E}^3$$

and

$$\tau_*(s) = \eta_*(s)\kappa_*(s), \quad s \in [0,L], \; t \in [-w,w]. \tag{3.6}$$

The deformation $\tilde{y}_*$ that takes each point $x_* \in \mathcal{D}_*$ to a point $y_* \in \mathcal{S}_* \subset \mathbb{E}^3$ is defined implicitly so that $y_*(s,t) = \tilde{y}_*(x_*(s,t))$ for each parameter pair $(s,t)$, $s \in [0,L]$, $t \in [-w,w]$, in conjunction with the assignment that the single closure point $r_*(0) = r_*(L)$ of the midline directrix in $\mathcal{S}_*$ corresponds to the two points $x_*(0,0) := o$ and $x_*(L,0) = L\imath_1$ in $\mathcal{D}_*$. Thus, the ends of the rectangular strip $\mathcal{D}_*$ in (3.6) are given, respectively, by $t\imath_2$ and $L\imath_1 + t\imath_2$ for $t \in [-w,w]$, which has the consequence that $\mathcal{D}_*$ and every $\mathcal{D}$ has the same overall rectangular geometric shape and dimensions.

It should be noted that Starostin & van der Heijden [6] & van der Heijden & Starostin [1] do not explicitly relate the parameter pairs $(s,t)$ in (3.1) to the *material points* of a *fixed undistorted* rectangular planar reference configuration and that they do not introduce the concept of an isometric deformation from such a *fixed undistorted* rectangular planar state to account for the unstretchable nature of the material. Thus, the unstretchability of an undistorted rectangular material strip is not addressed within their variational scheme.

A major missing element in the work of Starostin & van der Heijden [6] that remains missing in the clarification of van der Heijden & Starostin [1] is, as noted in §§1 and 2, the introduction of a fixed, undistorted reference configuration $\mathcal{R}$ for the rectangular material strip relative to which deformations of the material strip into a 180° twist Möbius band of the rectifying developable form (3.1) are to be measured. Unambiguously, the material points $z$ of $\mathcal{R}$ can be defined as

$$z = z(z_1, z_2) = z_1\imath_1 + z_2\imath_2, \quad z_1 \in [0.L], \; z_2 \in [-w,w], \tag{3.7}$$

together with a uniquely defined relationship between the parameter pairs $(s,t)$ in (3.1) and the parameter pairs $(z_1, z_2)$ in (3.7). Because the equilibrium shape $\mathcal{S}_*$ of (3.5) is considered to be fixed, albeit to be determined, within the set of smooth rectifying developable Möbius bands (3.1), and because the associated planar rectangular strip $\mathcal{D}_*$ of (3.5) has the same shape as $\mathcal{R}$, it is natural to determine $(s,t)$ in terms of $(z_1, z_2)$ through[3]

$$z_1 = s + t\eta_*(s) \quad \text{and} \quad z_2 = t, \quad s \in [0,L], \; t \in [-w,w], \tag{3.8}$$

and to consider the fixed, undistorted material reference configuration $\mathcal{R}$ for the rectangular material strip to be characterized in terms of $(s,t)$ as

$$z = (s + t\eta_*(s))\imath_1 + t\imath_2 \quad \text{and} \quad s \in [0,L], \; t \in [-w,w]. \tag{3.9}$$

With this characterization of $\mathcal{R}$, the mapping $z \in \mathcal{R} \mapsto y_* \in \mathcal{S}_*$ is a *isometric deformation* of the fixed, undistorted material reference configuration $\mathcal{R}$ to the rectifying developable equilibrium shape $\mathcal{S}_*$. But, it is important to observe that the mapping $z \in \mathcal{R} \mapsto y \in \mathcal{S}$, which is a *deformation* of the fixed, undistorted material reference configuration $\mathcal{R}$ to the rectifying developable shape $\mathcal{S}$ of (3.1), *is not an isometric deformation*. The consequence is that the set of rectifying developable Möbius bands as given in (3.1) that are considered to be competitors in the variational scheme of Starostin & van der Heijden [6], as augmented and clarified by van der Heijden & Starostin [1], do not satisfy the constraint embodying the unstretchability of the material.

---

[3]The inequality in Footnote 2 applies also to $\eta_*$.

## (c) Discussion

Starostin & van der Heijden [6] set up their investigation of the equilibrium shape of an unstretchable Möbius band by referencing the bending of a rectangular strip of paper because it 'deforms in such a way that its metrical properties are barely changed'. It is clear that by referring to 'its metrical properties', they had in mind *one* material object, and that their intention was to study the material length-preserving deformation of a given undistorted rectangular material strip into a free-standing 180° twist Möbius band and consequently characterize its equilibrium shape. They posed the problem as one in the calculus of variations wherein an appropriate bending energy functional, introduced above as the Wunderlich functional $\mathcal{W}$, was to be minimized over the set of all smooth rectifying developable 180° twist Möbius bands of given midline length $L$ and width $2w$, as introduced in (3.1). A fixed undistorted rectangular material reference configuration for the strip relative to which deformations to competitive surfaces (3.1) could be measured was not explicitly introduced. While it was introduced in the comments of van der Heijden & Starostin [1] as new information, it was not properly used in their equation (2), and the constraint of fixed 'metrical properties', namely material unstretchability, was not applied. Thus, the basic requirements constituting an appropriate and well-posed mechanics problem, for an elastic unstretchable solid surface, in the calculus of variations were left incomplete. Perhaps the unstated conditions were thought to be included by characterizing the competing surfaces in the variational strategy as the set of all smooth rectifying developable 180° twist Möbius bands of given midline length $L$ and width $2w$, because it is well-known that each surface in this set has a corresponding generator that is orthogonal to its midline directrix, which if severed can be rolled out onto, and developed into, a given flat rectangular shape of length $L$ and width $2w$. However, this observation only illustrates the differential geometric notion of an *isometry relation* between each Möbius band and a flat rectangle in $\mathbb{E}^2$. Moreover, as we explained in §§1 and 3b, while there is a specific isometric mapping of the rectangle itself to each specific rectifying developable Möbius band, the rectangle does not define a *fixed rectangular material reference configuration* relative to which a *deformation* to each Möbius band is measured.

By restricting the class of variational competitors to be the set of rectifying developable Möbius bands, and by neglecting to introduce a *fixed rectangular material reference configuration*, Starostin & van der Heijden [6], amended with their clarifying comments in van der Heijden & Starostin [1], did not address the major issues in the problem that they intended to solve. While, for fixed length $L$ and width $2w$ they determined through their variational scheme an optimal rectifying developable Möbius band $\mathcal{S}_*$, as is illustrated in (3.5), their variational scheme delivering $\mathcal{S}_*$ does not respect the requirement that the material be unstretchable. To emphasize this, we return to the fixed rectangular material reference configuration $\mathcal{R}$ of (3.7), relative to which all deformations are to be measured and objectively relate its material points to the parameters $(s, t)$ used in (3.1) to define the class of competitors $\mathcal{S}$. This is accomplished with the aid of (3.8) and (3.9). Thus, it follows that the deformation of the material points $z \in \mathcal{R} \mapsto y \in \mathcal{S}$ may be considered to be the composition of the deformation of the material points $z \in \mathcal{R} \mapsto x \in \mathcal{D}$ followed by the isometric mapping of the points $x \in \mathcal{D} \mapsto y \in \mathcal{S}$, as discussed in §3a. Clearly, from (3.2) and (3.9), the deformation of the material points $z \in \mathcal{R} \mapsto x \in \mathcal{D}$ may be written as

$$x - z = t(\eta(s) - \eta_*(s))\boldsymbol{\iota}_1 \quad \text{and} \quad s \in [0, L], \ t \in [-w, w], \tag{3.10}$$

where $(s, t)$ is determined in terms of $z$ through (3.7) and (3.8). Because $\eta(L) = -\eta(0) = \eta_*(L) = -\eta_*(0) = 0$, this deformation corresponds to an internal shear of the undistorted fixed rectangular material reference configuration $\mathcal{R}$.

Granted, Starostin & van der Heijden [6], amended with their clarifying comments in van der Heijden & Starostin [1], determine from their variational scheme a rectifying developable Möbius band $\mathcal{S}_*$ and a related flat rectangular reference configuration $\mathcal{D}_*$ such that $\mathcal{S}_*$ is the isometric deformation of $\mathcal{D}_*$. However, their variational scheme tolerates variations that support arbitrary material shearing, without penalty, and such admissibility is at odds with the nature of an unstretchable material surface. Thus, their work is related to the differential geometry of

rectifying developable surfaces rather than to the mechanics of unstretchable material surfaces. It identifies the Wunderlich [9,10] functional of (3.4), which is based in mechanics and represents the bending energy for rectifying developable surfaces, as a convenient dimensionally reduced functional to minimize over the class of rectifying developable surfaces and thereby obtain an optimal rectifying developable Möbius band of given length and width. From a mechanics point of view, the Euler–Lagrange equations determined from this minimization procedure do not express the equilibrium requirements in the mechanics of unstretchable ribbons because they are determined from a variational procedure that allows stretchable admissible variations.

The Wunderlich [9,10] functional is a correct dimensional reduction for accurately representing the bending energy of a free-standing, unknotted, non self-intersecting Möbius band by smoothly connecting together the two short ends of a given flat unstretchable rectangular material strip after a 180° twist. If the Möbius band is assumed to be represented as a rectifying developable surface $\mathcal{S}$, parameterized as in (3.1), then it is important that the end conditions $\eta(0) = 0$ and $\eta(L) = 0$ be satisfied—conditions not mentioned by Starostin & van der Heijden [6] but included as an amendment in van der Heijden & Starostin [1]. As noted earlier, the variational strategy proposed in [6], and as amended in [1], is to optimize the Wunderlich functional over all such rectifying developable Möbius bands. While the authors of [1] and [6] recognize that each competing band has the same planar development onto a rectangle of fixed length $L$ and width $2w$, they do not recognize that as a material surface each competing band they consider corresponds to a different rectangular reference configuration within the same fixed overall $L \times 2w$ rectangular shape. Relative to the one reference configuration that relates to their so-determined optimal Möbius band, all others are internally stretched and this is inconsistent with the hypothesis that the material is unstretchable.

In closing, we emphasize that any mechanics-based study of the isometric deformation of a given fixed rectangular material reference strip into a rectifying developable surface is of distinctly limited applicability. For example, the only way to maintain the parameterization of a rectifying developable and to isometrically deform a flat rectangular shape onto the surface of a right circular cylinder is to wrap it so that two of its parallel edges are coincident with the generators of the cylinder—but there are other ways, helical in form, that do not involve maintaining the parameterization of a rectifying developable, to achieve an isometric wrapping. Moreover, as Chen, Fosdick & Fried [4, §4.3] showed, it is not even possible to isometrically deform a given rectangular material reference strip onto the surface of a right circular cone while complying with the parameterization of a rectifying developable. But, of course, a rectangular reference strip can be so deformed.

Acknowledgement. EF gratefully acknowledges support from the Okinawa Institute of Science and Technology Graduate University with subsidy funding from the Cabinet Office, Government of Japan.

Data accessibility. This article has no additional data.

Authors' contributions. Y.-C.C: conceptualization, formal analysis, investigation, writing—original draft, writing—review and editing; R.F.: conceptualization, formal analysis, investigation, writing—original draft, writing—review and editing; E.F.: conceptualization, formal analysis, investigation, writing—original draft, writing—review and editing.

All authors gave final approval for publication and agreed to be held accountable for the work performed therein.

Conflict of interest declaration. We declare that we have no competing interests.

Funding. No funding has been received for this article.

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
