## [Peer Review File · Proceedings. Mathematical, Physical, and Engineering Sciences]

Review History

RSPA-2021-0856.R0 (Original submission)

Review form: Referee 1

Is the manuscript an original and important contribution to its field?

Acceptable

Is the paper of sufficient general interest?

Marginal

Is the overall quality of the paper suitable?

Acceptable

Can the paper be shortened without overall detriment to the main message?

Yes

Do you think some of the material would be more appropriate as an electronic appendix?

No

Do you have any ethical concerns with this paper?

No

Recommendation?

Accept as is

Comments to the Author(s)

Happy to see this response published more-or-less as is.

Review form: Referee 2

Is the manuscript an original and important contribution to its field?

Good

Is the paper of sufficient general interest?

Good

Is the overall quality of the paper suitable?

Good

Can the paper be shortened without overall detriment to the main message?

No

Do you think some of the material would be more appropriate as an electronic appendix?

No

Do you have any ethical concerns with this paper?

No

Recommendation?

Major revision is needed (please make suggestions in comments)

Comments to the Author(s)

I have been asked to referee

[1] "RSPA-2021-0856 - "Reply to the comment of van der Heijden and Starostin"

where the comment in question is

[2] RSPA-2021-0629.R1 which I have not explicitly been asked to comment on.

Note that paper [2] is comment on

[3] Chen Y-C, Fried E. 2016 Möbius bands, unstretchable material sheets and developable surfaces. Proc. R. Soc. A 472: 20160459,

which the authors of [2] claim implicitly criticizes their previous work which I shall call

[4] Starostin EL, van der Heijden GHM. 2007 The shape of a Möbius strip. Nat. Mater. 6, 563–567.

I feel that the correspondence is potentially interesting and deserves to be published. But this reply [1] to the comment [2] cannot be published as is. I hope that it is not too late to ask the authors of [2] to make some changes too. But I feel the reply [1] as it stands is not publishable. I

set out my reasons below along with a proposed resolution.

Major points

1.

The Rhetoric of both [2] and especially [1] needs to be toned down.

Both works use words like:

"explain in more detail precisely where Chen & Fried go wrong" [2]

"purported proof" [2]

"all [their papers] are based on a misunderstanding" [2]

"grossly distorts" [1]

"egregious criticisms" [1]

"fatal mistake" [1]

"The inconsistency is unacceptable" [1]

"The authors did not even confront the problem they intended to solve"

etc.

2.

Chen and Fried claim false pique rather than address the specific criticism in [2]. Perhaps this is understandable because van der Heijden and Starostin start with a claim that Chen and Fred are wrong. Clearly both parties are upset. But this upset should not make it into scholarly publications. Chen and Fried start with the assertion that

"Chen and Fried ([3]) do not state that the solution of Starostin and van der Heijden [4] is incorrect"

This is at best debatable, because, if you read even the abstract of [3], it is clear that the authors categorise a vast swathe of previous research on this topic as being "erroneous". I feel that if Starostin and van der Heijden were to rewrite a few of the opening sentences of their comment this might help. For example if they opened with.

"We note the enhanced variational formulation of the problem of finding equilibrium surfaces of developable surfaces proposed by Chen and Fried in their paper [3]. We also note that the authors claim that earlier work on finding equilibrium shapes of Mobius strips, including our own prior work [4] is "erroneous. While we don't deny there is scientific merit in the enhanced variational formulations proposed by Chen, Fried and their collaborators in their later papers [*,*,*] we should like to point out that the theory we presented in our prior work is self-consistent and indeed we are not aware of any claims that the equilibrium shape we have found is substantially incorrect ..."

Then, instead of saying "where Chen and Fred go wrong" and phrases like that, it would be better to refer only to why your solution is consistent with their formulation if one imposes constraints via the boundary conditions.

Then, I feel Chen and Fried should confine their reply only to whether or not they agree with the claims in [2]. Most of the current version of [1] seems to be more a justification of [3], repeating many of its arguments, while coming up with many new criticisms of [4]. I feel they should try to

write a revised reply to the comment that is no longer than the comment itself.

3.

Large parts of [1] I find very hard to read. There is a lot here in their justification that I would class as "rational mechanics". Unfortunately from a philosophical point of view, pure rational mechanics does not exist in my view; all theories that purport to describe the real world necessarily have inconsistencies, especially theories that involve thin plates rather than full 3D elasticity theory. Their paper is full of justifications of their position "from the principles of mechanics". Just as Euler-Bernoulli and Timoshenko beams contain inconsistencies, so to the Kirchoff-style models of bending and stretching of a flat plate.

Where [1] and indeed [3], on which it is based, is on strong ground is in the authors' arguments based on the the calculus of variations (which again, when applied to mechanics is just a model). The main point of [3] and indeed their subsequent papers is to show that theories of deformable surfaces need to carefully consider the space of allowable deformations in order to make mathematically rigorous statements. They show that the work [4] and others like it could be extended by considering other variational formulations that are in some sense more consistent (never completely consistent, in my view) and would allow a different set of test functions for the variational principle. However they stop short of actually posing such problems and solving them in the case of the Mobius strip.

Proposed resolution:

My feeling here is that the resolution is subtle. I don't believe it is a question of either of the papers [3] or [4] being "wrong". In my view [4] is a calculation that makes certain implicit assumptions, whereas [3] is a paper that would propose a more general theory to problems like this, which *may* lead to solutions that are more consistent with reality.

Chen and Fried claim in [1] that paper [3] was not intended to say that [4] (and other papers like it) are incorrect. Nevertheless van der Heijden and Starostin, the authors of [4], have taken exception to the word "erroneous" (and others like it) within [3]. This has led them to claim in [2] that Chen and Fried are wrong, which has led to an escalation.

My proposed resolution is a mild toning down of [2] to withdraw the claim that Chen and Fried are wrong and then for Chen and Fried to be invited to write a much more succinct version of their reply [1] that takes on board the fact that van der Heijden and Starostin are no longer claiming any errors in [1], but are simply pointing out that their theory is not as flawed as a reader of [1] might think.

Review form: Referee 3

Is the manuscript an original and important contribution to its field?

Good

Is the paper of sufficient general interest?

Good

Is the overall quality of the paper suitable?

Good

Do you think some of the material would be more appropriate as an electronic appendix?

No

Do you have any ethical concerns with this paper?

Yes

Recommendation?

Major revision is needed (please make suggestions in comments)

Comments to the Author(s)

Please see Appendix A.

Decision letter (RSPA-2021-0856.R0)

10-Jan-2022

Dear Professor Fried

The Editor of Proceedings A has now received comments from referees on the above paper and would like you to revise it in accordance with their suggestions which can be found below (not including confidential reports to the Editor).

Please submit a copy of your revised paper within four weeks - if we do not hear from you within this time then it will be assumed that the paper has been withdrawn. In exceptional circumstances, extensions may be possible if agreed with the Editorial Office in advance.

Please note that it is the editorial policy of Proceedings A to offer authors one round of revision in which to address changes requested by referees. If the revisions are not considered satisfactory by the Editor, then the paper will be rejected, and not considered further for publication by the journal. In the event that the author chooses not to address a referee's comments, and no scientific justification is included in their cover letter for this omission, it is at the discretion of the Editor whether to continue considering the manuscript.

To revise your manuscript, log into <http://mc.manuscriptcentral.com/prsa> and enter your Author Centre, where you will find your manuscript title listed under "Manuscripts with Decisions." Under "Actions," click on "Create a Revision." Your manuscript number has been appended to denote a revision.

You will be unable to make your revisions on the originally submitted version of the manuscript. Instead, revise your manuscript and upload a new version through your Author Centre.

When submitting your revised manuscript, you will be able to respond to the comments made by the referee(s) and upload a file "Response to Referees" in Step 1: "View and Respond to Decision Letter". Please provide a point-by-point response to the comments raised by the reviewers and the editor(s). A thorough response to these points will help us to assess your revision quickly. You can also upload a 'tracked changes' version either as part of the 'Response to reviews' or as a 'Main document'.

IMPORTANT: Your original files are available to you when you upload your revised manuscript. Please delete any unnecessary previous files before uploading your revised version.

When revising your paper please ensure that it remains under 28 pages long. In addition, any pages over 20 will be subject to a charge (£150 + VAT (where applicable) per page). Your paper has been ESTIMATED to be 10 pages.

Open Access

You are invited to opt for open access, our author pays publishing model. Payment of open access fees will enable your article to be made freely available via the Royal Society website as soon as it is ready for publication. For more information about open access please visit <https://royalsociety.org/journals/authors/open-access/>. The open access fee for this journal is £1700/\$2380/€2040 per article. VAT will be charged where applicable. Please note that if the corresponding author is at an institution that is part of a Read and Publishing deal you are required to select this option. See <https://royalsociety.org/journals/librarians/purchasing/read-and-publish/read-publish-agreements/> for further details.

Once again, thank you for submitting your manuscript to Proc. R. Soc. A and I look forward to receiving your revision. If you have any questions at all, please do not hesitate to get in touch.

Yours sincerely
Raminder Shergill
proceedingsa@royalsociety.org

Reviewer(s)' Comments to Author:

Referee: 1

Comments to the Author(s)

Happy to see this response published more-or-less as is.

Referee: 2

Comments to the Author(s)

I have been asked to referee

[1] "RSPA-2021-0856 - "Reply to the comment of van der Heijden and Starostin"

where the comment in question is

[2] RSPA-2021-0629.R1 which I have not explicitly been asked to comment on.

Note that paper [2] is comment on

[3] Chen Y-C, Fried E. 2016 Möbius bands, unstretchable material sheets and developable surfaces. Proc. R. Soc. A 472: 20160459,

which the authors of [2] claim implicitly criticizes their previous work which I shall call

[4] Starostin EL, van der Heijden GHM. 2007 The shape of a Möbius strip. Nat. Mater. 6, 563–567.

I feel that the correspondence is potentially interesting and deserves to be published. But this reply [1] to the comment [2] cannot be published as is. I hope that it is not too late to ask the authors of [2] to make some changes too. But I feel the reply [1] as it stands is not publishable. I set out my reasons below along with a proposed resolution.

Major points

1.

The Rhetoric of both [2] and especially [1] needs to be toned down.

Both works use words like:

"explain in more detail precisely where Chen & Fried go wrong" [2]

"purported proof" [2]

"all [their papers] are based on a misunderstanding" [2]

"grossly distorts" [1]

"egregious criticisms" [1]

"fatal mistake" [1]

"The inconsistency is unacceptable" [1]

"The authors did not even confront the problem they intended to solve"

etc.

2.

Chen and Fried claim false pique rather than address the specific criticism in [2]. Perhaps this is understandable because van der Heijden and Starostin start with a claim that Chen and Fred are wrong. Clearly both parties are upset. But this upset should not make it into scholarly publications. Chen and Fried start with the assertion that

"Chen and Fried ([3]) do not state that the solution of Starostin and van der Heijden [4] is incorrect"

This is at best debatable, because, if you read even the abstract of [3], it is clear that the authors categorise a vast swathe of previous research on this topic as being "erroneous". I feel that if Starostin and van der Heijden were to rewrite a few of the opening sentences of their comment this might help. For example if they opened with.

"We note the enhanced variational formulation of the problem of finding equilibrium surfaces of developable surfaces proposed by Chen and Fried in their paper [3]. We also note that the authors claim that earlier work on finding equilibrium shapes of Mobius strips, including our own prior work [4] is "erroneous. While we don't deny there is scientific merit in the enhanced variational formulations proposed by Chen, Fried and their collaborators in their later papers [*,*,*] we should like to point out that the theory we presented in our prior work is self-consistent and indeed we are not aware of any claims that the equilibrium shape we have found is substantially incorrect ..."

Then, instead of saying "where Chen and Fred go wrong" and phrases like that, it would be better to refer only to why your solution is consistent with their formulation if one imposes constraints via the boundary conditions.

Then, I feel Chen and Fried should confine their reply only to whether or not they agree with the claims in [2]. Most of the current version of [1] seems to be more a justification of [3], repeating many of its arguments, while coming up with many new criticisms of [4]. I feel they should try to write a revised reply to the comment that is no longer than the comment itself.

3.

Large parts of [1] I find very hard to read. There is a lot here in their justification that I would class as "rational mechanics". Unfortunately from a philosophical point of view, pure rational

mechanics does not exist in my view; all theories that purport to describe the real world necessarily have inconsistencies, especially theories that involve thin plates rather than full 3D elasticity theory. Their paper is full of justifications of their position "from the principles of mechanics". Just as Euler-Bernoulli and Timoshenko beams contain inconsistencies, so to the Kirchoff-style models of bending and stretching of a flat plate.

Where [1] and indeed [3], on which it is based, is on strong ground is in the authors' arguments based on the the calculus of variations (which again, when applied to mechanics is just a model). The main point of [3] and indeed their subsequent papers is to show that theories of deformable surfaces need to carefully consider the space of allowable deformations in order to make mathematically rigorous statements. They show that the work [4] and others like it could be extended by considering other variational formulations that are in some sense more consistent (never completely consistent, in my view) and would allow a different set of test functions for the variational principle. However they stop short of actually posing such problems and solving them in the case of the Mobius strip.

Proposed resolution:

My feeling here is that the resolution is subtle. I don't believe it is a question of either of the papers [3] or [4] being "wrong". In my view [4] is a calculation that makes certain implicit assumptions, whereas [3] is a paper that would propose a more general theory to problems like this, which *may* lead to solutions that are more consistent with reality.

Chen and Fried claim in [1] that paper [3] was not intended to say that [4] (and other papers like it) are incorrect. Nevertheless van der Heijden and Starostin, the authors of [4], have taken exception to the word "erroneous" (and others like it) within [3]. This has led them to claim in [2] that Chen and Fried are wrong, which has led to an escalation.

My proposed resolution is a mild toning down of [2] to withdraw the claim that Chen and Fried are wrong and then for Chen and Fried to be invited to write a much more succinct version of their reply [1] that takes on board the fact that van der Heijden and Starostin are no longer claiming any errors in [1], but are simply pointing out that their theory is not as flawed as a reader of [1] might think.

Referee: 3

Comments to the Author(s)

Please see Appendix A.

Author's Response to Decision Letter for (RSPA-2021-0856.R0)

See Appendix B.

RSPA-2021-0856.R1 (Revision)

Review form: Referee 1

Is the manuscript an original and important contribution to its field?

Good

Is the paper of sufficient general interest?

Good

Is the overall quality of the paper suitable?

Good

Can the paper be shortened without overall detriment to the main message?

Yes

Do you think some of the material would be more appropriate as an electronic appendix?

No

Do you have any ethical concerns with this paper?

No

Recommendation?

Accept as is

Comments to the Author(s)

Authors appear to have answered the issues adequately and I am happy that the paper is acceptable as is.

Review form: Referee 2

Is the manuscript an original and important contribution to its field?

Good

Is the paper of sufficient general interest?

Good

Is the overall quality of the paper suitable?

Good

Can the paper be shortened without overall detriment to the main message?

Yes

Do you think some of the material would be more appropriate as an electronic appendix?

No

Do you have any ethical concerns with this paper?

No

Recommendation?

Accept with minor revision (please list in comments)

Comments to the Author(s)

I thank the authors for toning down the rhetoric as I suggested. I have asked the editor if the authors of the original comment can also be asked to tone down their rhetoric too by making some further minor changes at the proof stage.

There is still one major point I disagree with... It is the sentence "Chen and Fried [2] do not state that the solution of Starostin and van der Heijden [6] is incorrect". I disagree with the authors' assertion that the authors state in their reply to my review that they did not state this even implicitly. Interpretation and meaning of phrases is something that is primarily at the discretion of the reader, not the author. I certainly got the impression that the word "erroneous" when referring to formulations which include that in [6] meant that the authors of [2] were implicitly saying that the solution in [6] is "wrong". I therefore ask the authors to rephrase this part of their response... something like

We appreciate that Starostin and van der Heijden have formed the impression that the authors of [2] were claiming that the solution of [6] is incorrect. We should like to point out though that this was not explicitly stated in [2], the intention of which was to explore whether...

Review form: Referee 3

Is the manuscript an original and important contribution to its field?

Marginal

Is the paper of sufficient general interest?

Marginal

Is the overall quality of the paper suitable?

Marginal

Can the paper be shortened without overall detriment to the main message?

No

Do you think some of the material would be more appropriate as an electronic appendix?

No

Do you have any ethical concerns with this paper?

Yes

Recommendation?

Reject – article is scientifically unsound

Comments to the Author(s)

In their response to the referee reports, the authors write that "at best, [Starostin and van der Heijden] solve the differential geometry problem of finding the Möbius band that optimizes the Wunderlich functional among all rectifying developable Möbius bands that correspond to the planar development of a rectangle of length L and width $2w$ ". This is exactly what I have always thought was Starostin and van der Heijden's contribution. (I agree that they could have added more information when explaining the method.)

In my opinion, the authors' statement is a rather curious admission: in "Möbius bands, unstretchable material sheets and developable surfaces", Chen and Fried criticizes earlier work on developable surfaces without contemplating the possibility that some of it might not be addressing the authors' intended mechanical problem, but rather the geometric one (which I suppose should deserve some respect too).

As a matter of fact, the distinction between these two problems remains a complete mystery to me. The authors insist that, from the point of view of mechanics, Starostin and van der Heijden's approach does not lead to the correct solution. They note that, albeit any two different surfaces competing in the variational scheme have the same planar development, it is not possible to map isometrically one to the other through their "rectifying parametrizations". I fully agree with this observation, but I reject their conclusion, as the bending energy does not depend on the parametrization, only on its image. Hence, I do not see why the lack of a fixed reference configuration should be an issue.

Decision letter (RSPA-2021-0856.R1)

21-Mar-2022

Dear Professor Fried,

On behalf of the Editor, I am pleased to inform you that your Manuscript RSPA-2021-0856.R1 entitled "Reply to the comment of van der Heijden and Starostin" has been accepted for publication subject to minor revisions in Proceedings A. Please find the referees' comments below.

The reviewer(s) have recommended publication, but also suggest some minor revisions to your manuscript. Therefore, I invite you to respond to the reviewer(s)' comments and revise your manuscript. Please note that we have a strict upper limit of 28 pages for each paper. Please endeavour to incorporate any revisions while keeping the paper within journal limits. Please note that page charges are made on all papers longer than 20 pages. If you cannot pay these charges you must reduce your paper to 20 pages before submitting your revision. Your paper has been ESTIMATED to be 10 pages. We cannot proceed with typesetting your paper without your agreement to meet page charges in full should the paper exceed 20 pages when typeset. If you have any questions, please do get in touch.

It is a condition of publication that you submit the revised version of your manuscript within 7 days. If you do not think you will be able to meet this date please let me know in advance of the due date.

To revise your manuscript, log into <https://mc.manuscriptcentral.com/prsa> and enter your Author Centre, where you will find your manuscript title listed under "Manuscripts with Decisions." Under "Actions," click on "Create a Revision." Your manuscript number has been appended to denote a revision.

You will be unable to make your revisions on the originally submitted version of the manuscript. Instead, revise your manuscript and upload a new version through your Author Centre.

When submitting your revised manuscript, you will be able to respond to the comments made by the referee(s) and upload a file "Response to Referees" in Step 1: "View and Respond to Decision Letter". Please provide a point-by-point response to the comments raised by the reviewers and the editor(s). A thorough response to these points will help us to assess your revision quickly. You can also upload a 'tracked changes' version either as part of the 'Response to reviews' or as a 'Main document'.

IMPORTANT: Your original files are available to you when you upload your revised manuscript. Please delete any redundant files before completing the submission process.

When uploading your revised files, please make sure that you include the following as we cannot proceed without these:

- 1) A text file of the manuscript (doc, txt, rtf or tex), including the references, tables (including captions) and figure captions. Please remove any tracked changes from the text before submission. PDF files are not an accepted format for the "Main Document".
- 2) A separate electronic file of each figure (tif, eps or print-quality pdf preferred). The format should be produced directly from original creation package, or original software format.
- 3) Electronic Supplementary Material (ESM): all supplementary materials accompanying an accepted article will be treated as in their final form. Note that the Royal Society will not edit or typeset supplementary material and it will be hosted as provided. Please ensure that the supplementary material includes the paper details where possible (authors, article title, journal name). Supplementary files will be published alongside the paper on the journal website and posted on the online figshare repository (<https://figshare.com>). The heading and legend provided for each supplementary file during the submission process will be used to create the figshare page, so please ensure these are accurate and informative so that your files can be found in searches. Files on figshare will be made available approximately one week before the accompanying article so that the supplementary material can be attributed a unique DOI.

Alternatively you may upload a zip folder containing all source files for your manuscript as described above with a PDF as your "Main Document". This should be the full paper as it appears when compiled from the individual files supplied in the zip folder.

Article Funder

Please ensure you fill in the Article Funder question on page 2 to ensure the correct data is collected for FundRef (<http://www.crossref.org/fundref/>).

Media summary

Please ensure you include a short non-technical summary (up to 100 words) of the key findings/importance of your paper. This will be used for to promote your work and marketing purposes (e.g. press releases). The summary should be prepared using the following guidelines:

- *Write simple English: this is intended for the general public. Please explain any essential technical terms in a short and simple manner.
- *Describe (a) the study (b) its key findings and (c) its implications.
- *State why this work is newsworthy, be concise and do not overstate (true 'breakthroughs' are a rarity).
- *Ensure that you include valid contact details for the lead author (institutional address, email address, telephone number).

Cover images

We welcome submissions of images for possible use on the cover of Proceedings A. Images should be square in dimension and please ensure that you obtain all relevant copyright permissions before submitting the image to us. If you would like to submit an image for consideration please send your image to proceedingsa@royalsociety.org

Open Access

You are invited to opt for open access, our author pays publishing model. Payment of open access fees will enable your article to be made freely available via the Royal Society website as soon as it is ready for publication. For more information about open access please visit <https://royalsociety.org/journals/authors/open-access/>. The open access fee for this journal is £1700/\$2380/€2040 per article. VAT will be charged where applicable. Please note that if the corresponding author is at an institution that is part of a Read and Publishing deal you are required to select this option. See <https://royalsociety.org/journals/librarians/purchasing/read-and-publish/read-publish-agreements/> for further details.

Once again, thank you for submitting your manuscript to Proceedings A and I look forward to receiving your revision. If you have any questions at all, please do not hesitate to get in touch.

Best wishes
 Raminder Shergill
proceedingsa@royalsociety.org
 Proceedings A

Reviewer(s)' Comments to Author:

Referee: 2

Comments to the Author(s)

I thank the authors for toning down the rhetoric as I suggested. I have asked the editor if the authors of the original comment can also be asked to tone down their rhetoric too by making some further minor changes at the proof stage.

There is still one major point I disagree with... It is the sentence "Chen and Fried [2] do not state that the solution of Starostin and van der Heijden [6] is incorrect". I disagree with the authors assertion that the authors state in their reply to my review that they did not state this even implicitly. Interpretation and meaning of phrases is something that is primarily at the discretion of the reader, not the author. I certainly got the impression that the word "erroneous" when referring to formulations which include that in [6] meant that the authors of [2] were implicitly saying that the solution in [6] is "wrong". I therefore ask the authors to rephrase this part of their response... something like

We appreciate that Starostin and van der Heijden have formed the impression that the authors of [2] were claiming that the solution of [6] is incorrect. We should like to point out though that this was not explicitly stated in [2], the intention of which was to explore whether...

Referee: 1

Comments to the Author(s)

Authors appear to have answered the issues adequately and I am happy that the paper is acceptable as is.

Referee: 3

Comments to the Author(s)

In their response to the referee reports, the authors write that "at best, [Starostin and van der Heijden] solve the differential geometry problem of finding the Möbius band that optimizes the Wunderlich functional among all rectifying developable Möbius bands that correspond to the planar development of a rectangle of length L and width $2w$ ". This is exactly what I have always thought was Starostin and van der Heijden's contribution. (I agree that they could have added more information when explaining the method.)

In my opinion, the authors' statement is a rather curious admission: in "Möbius bands, unstretchable material sheets and developable surfaces", Chen and Fried criticizes earlier work on developable surfaces without contemplating the possibility that some of it might not be addressing the authors' intended mechanical problem, but rather the geometric one (which I suppose should deserve some respect too).

As a matter of fact, the distinction between these two problems remains a complete mystery to me. The authors insist that, from the point of view of mechanics, Starostin and van der Heijden's approach does not lead to the correct solution. They note that, albeit any two different surfaces competing in the variational scheme have the same planar development, it is not possible to map isometrically one to the other through their "rectifying parametrizations". I fully agree with this observation, but I reject their conclusion, as the bending energy does not depend on the parametrization, only on its image. Hence, I do not see why the lack of a fixed reference configuration should be an issue.

Author's Response to Decision Letter for (RSPA-2021-0856.R1)

See Appendix C.

Decision letter (RSPA-2021-0856.R2)

@@date to be populated upon sending@@

Dear Colleagues,

I am writing to inform you that the Editor has made a decision on manuscript entitled "Reply to the comment of van der Heijden and Starostin" which you kindly refereed for Proceedings A. Please find the authors' decision letter below.

On behalf of the Editor of Proceedings A, we thank you for your help with this article and we look forward to your input in the future.

Decision made on this manuscript: Accept as is

Best wishes
Raminder Shergill
proceedingsa@royalsociety.org

@@date to be populated upon sending@@

Dear Professor Fried

On behalf of the Editor, I am pleased to inform you that your manuscript entitled "Reply to the comment of van der Heijden and Starostin" has been accepted in its final form for publication in Proceedings A.

Our Production Office will be in contact with you in due course. You can expect to receive a proof of your article soon. Please contact the office to let us know if you are likely to be away from e-mail in the near future. If you do not notify us and comments are not received within 5 days of sending the proof, we may publish the paper as it stands.

As a reminder, you have provided the following 'Data accessibility statement' (if applicable). Please remember to make any data sets live prior to publication, and update any links as needed when you receive a proof to check. It is good practice to also add data sets to your reference list.
Statement (if applicable):

Under the terms of our licence to publish you may post the author generated postprint (ie. your accepted version not the final typeset version) of your manuscript at any time and this can be made freely available. Postprints can be deposited on a personal or institutional website, or a recognised server/repository. Please note however, that the reporting of postprints is subject to a media embargo, and that the status the manuscript should be made clear. Upon publication of the definitive version on the publisher's site, full details and a link should be added.

You can cite the article in advance of publication using its DOI. The DOI will take the form: 10.1098/rspa.XXXX.YYYY, where XXXX and YYYY are the last 8 digits of your manuscript number (eg. if your manuscript number is RSPA-2017-1234 the DOI would be 10.1098/rspa.2017.1234).

For tips on promoting your accepted paper see our blog post:
<https://royalsociety.org/blog/2020/07/promoting-your-latest-paper-and-tracking-your-results/>

Thank you for your submission. On behalf of the Editors of the journal, we look forward to your continued contributions to the Journal.

Best wishes
Raminder Shergill,
Proceedings A Editorial Office
proceedingsa@royalsociety.org

Appendix A

Comments to the authors

In its current form, I do not deem the paper suitable for publication in Proceedings A. The reason is that the authors' critique of Starostin and van der Heijden's variational approach is far from being clear and neatly explained.

The problem at issue is purely geometrical. It is clear that, in general, the standard parametrization of a developable surface does not represent the isometry mapping the surface to its planar development; equivalently, such parametrization is not an isometric embedding. However, one can use *any* parametrization to compute the bending energy (defined as the integral of the mean curvature squared).

More precisely, suppose that \mathcal{S} is a developable Möbius strip and let \mathcal{D} be the planar development (i.e., isometric image) of \mathcal{S} . The problem is "merely" to find, among all developable Möbius strips having the same development \mathcal{D} , the one, say \mathcal{S}_* , that minimizes the bending energy. As the energy is invariant under reparametrization, the choice of parametrization employed in the process is completely irrelevant. Once we have \mathcal{S}_* , we know that there exists an isometry from \mathcal{S}_* to \mathcal{D} , and so we are done.

It is therefore completely unclear to me how the authors are able to claim that Starostin and van der Heijden's variational approach does not give the correct solution. Note that, in their comment, Starostin and van der Heijden are not speaking about the variational scheme; they are only describing the isometry between a given \mathcal{S} and \mathcal{D} .

As far as I understand, the desired fixed "reference configuration" is just the set \mathcal{D} , which has zero energy (in any coordinate system). The authors seem to have in mind two distinct problems: one "related to the differential geometry of rectifying developable surfaces"; the other "to the mechanics of unstretchable material surfaces". It would be nice if the authors could explain how these two problems are different.

In conclusion, I suggest that the authors carefully reconsider their critique. In my opinion, to help the reader understand the authors' point of view, the paper should be less verbose and a bit more rigorous from a mathematical viewpoint.

Appendix B

Response to the reviews of RSPA-2021-0856

Yi-chao Chen, Roger Fosdick and Eliot Fried

Response to Reviewer #1

Happy to see this response published more-or-less as is.

We appreciate the supportive statement from Reviewer #1.

Response to Reviewer #2

I have been asked to referee [1] “RSPA-2021-0856 – “Reply to the comment of van der Heijden and Starostin”, where the comment in question is [2] RSPA-2021-0629.R1 which I have not explicitly been asked to comment on. Note that paper [2] is comment on [3] Chen Y-C, Fried E. 2016 Möbius bands, unstretchable material sheets and developable surfaces. Proc. R. Soc. A 472: 20160459, which the authors of [2] claim implicitly criticizes their previous work which I shall call [4] Starostin EL, van der Heijden GHM. 2007 The shape of a Möbius strip. Nat. Mater. 6, 563–567.

I feel that the correspondence is potentially interesting and deserves to be published. But this reply [1] to the comment [2] cannot be published as is. I hope that it is not too late to ask the authors of [2] to make some changes too. But I feel the reply [1] as it stands is not publishable. I set out my reasons below along with a proposed resolution.

We are grateful that Reviewer #2 finds our work to be potentially interesting and deserves to be published, albeit subject to revisions. We have substantially revised our presentation in response to the remarks of Reviewer #2. Our replies to the three “Major points” of the review are listed below, wherein [1], [2], and [6] refer to the corresponding references in our revision of RSPA-2021-0856.

Major point 1

The rhetoric of both [2] and especially [1] needs to be toned down. Both works use words like:

“explain in more detail precisely where Chen & Fried go wrong” [2]

“purported proof” [2]

“all [their papers] are based on a misunderstanding” [2]

“grossly distorts” [1]

“egregious criticisms” [1]

“fatal mistake” [1]

“The inconsistency is unacceptable” [1]

“The authors did not even confront the problem they intended to solve” etc.

Per the suggestion of Reviewer #2, we have conscientiously toned down our rhetoric while ensuring that our reply is based on objective statements and principled reasoning.

Chen and Fried claim false pique rather than address the specific criticism in [2]. Perhaps this is understandable because van der Heijden and Starostin start with a claim that Chen and Fred are wrong. Clearly both parties are upset. But this upset should not make it into scholarly publications. Chen and Fried ([3]) start with the assertion that “Chen and Fried [3] do not state that the solution of Starostin and van der Heijden [4] is incorrect”. This is at best debatable, because, if you read even the abstract of [3], it is clear that the authors categorise a vast swathe of previous research on this topic as being “erroneous”.

We agree and appreciate the comment “But this upset should not make it into scholarly publications.” As noted above, we have modified our submission to remove all incendiary language.

The assertion “Chen and Fried [3] do not state that the solution of Starostin and van der Heijden [4] is incorrect” is intended as a direct, word-to-word response to van der Heijden and Starostin’s statement in [2] that “Chen & Fried [3] claim that our solution of the shape of a material Möbius strip (taken to be inextensible) in [4] is incorrect. . .”, the purpose being to clarify that we did not make such a claim, either explicitly or implicitly. In the statement “the notion of a rectifying developable cannot be used to describe a pure bending of a rectangular region into a Möbius band or a generic ribbon, as has been erroneously done in many publications” from the abstract of [3] we sought to point out that a rectifying developable surface, as represented by a parametric form appearing in many publications, cannot generally be isometrically mapped to a rectangular planar strip. The example in Section 8 of [3] establishes this fact. For a Möbius band, supplemental conditions are required to ensure that a rectifying developable parametrization covers the band. Those conditions did not appear in [6]. However, they have been supplied, after the fact, in [1]. We did not single out [4], even though the first sentence of [4] reads “The Möbius strip, obtained by taking a rectangular strip of plastic or paper. . .” and the first equation of [4] gives the parametric form of a rectifying developable surface, which, as we demonstrate with an explicit example in [3] cannot generally be isometrically mapped to a rectangular strip.

I feel that if Starostin and van der Heijden were to rewrite a few of the opening sentences of their comment this might help. For example if they opened with:

“We note the enhanced variational formulation of the problem of finding equilibrium surfaces of developable surfaces proposed by Chen and Fried in their paper [3]. We also note that the authors claim that earlier work on finding equilibrium shapes of Möbius strips, including our own prior work [4] is ”erroneous. While we don’t deny there is scientific merit in the enhanced variational formulations proposed by Chen, Fried and their collaborators in their later papers [*,*,*,*] we should like to point out that the theory we presented in our prior work is self-consistent and indeed we are not aware of any claims that the equilibrium shape we have found is substantially incorrect. . .”

Then, instead of saying “where Chen and Fried go wrong” and phrases like that, it would be better to refer only to why your solution is consistent with their formulation if one imposes constraints via the boundary conditions.

We appreciate the Reviewer’s proposed revision to the opening sentences of the comment. However, our understanding is that [1] has been accepted for publication in its present form.

Then, I feel Chen and Fried should confine their reply only to whether or not they agree with the claims in [2]. Most of the current version of [1] seems to be more a justification of [3], repeating many of its arguments, while coming up with many new criticisms of [4]. I feel they should try to write a revised reply to the comment that is no longer than the comment itself.

We have made a concerted effort to keep our response as brief as possible. The comment [1] contains new information and claims that did not appear in [6].

We have addressed the specific comments of van der Heijden and Starostin [1] and have given a “reason based” reply to their major criticisms. We have responded to the claims and arguments of van der Heijden and Starostin [1] with statements based very specifically on what they wrote. Whenever our point of view differs from theirs, we have provided reasons why.

It is of course our responsibility to respond when van der Heijden and Starostin [1] make claims about our work that are unjustified, or cast doubt about our results or our research goals. But,

again, we do so in a professional, non-confrontal, and responsible manner that should not create “heat”.

We do not agree with the strategy promoted by van der Heijden and Starostin [1] and we give reasons why. We respond specifically to the steps that they lay out in their variational procedure and we provide a basis for why their approach is in contradiction with the unstretchability of the material.

Our response definitely is not based upon justifying our work by simply repeating our published arguments and, in so doing, coming up with new criticisms of their work, as Reviewer #2 suggested. In Section 3 of our response, we give a brief, but detailed, unpublished justification of our response in order to give a clear explanation and basis for the position we have taken.

Major point 2

Large parts of [1] I find very hard to read. There is a lot here in their justification that I would class as “rational mechanics”. Unfortunately from a philosophical point of view, pure rational mechanics does not exist in my view; all theories that purport to describe the real world necessarily have inconsistencies, especially theories that involve thin plates rather than full 3D elasticity theory. Their paper is full of justifications of their position “from the principles of mechanics”. Just as Euler–Bernoulli and Timoshenko beams contain inconsistencies, so to the Kirchhoff–style models of bending and stretching of a flat plate.

Where [1] and indeed [3], on which it is based, is on strong ground is in the authors’ arguments based on the the calculus of variations (which again, when applied to mechanics is just a model). The main point of [3] and indeed their subsequent papers is to show that theories of deformable surfaces need to carefully consider the space of allowable deformations in order to make mathematically rigorous statements. They show that the work [4] and others like it could be extended by considering other variational formulations that are in some sense more consistent (never completely consistent, in my view) and would allow a different set of test functions for the variational principle. However they stop short of actually posing such problems and solving them in the case of the Möbius strip.

Our reply to van der Heijden and Starostin [1] is based on broadly accepted understandings of kinematics, mechanics, and variational calculus. The problem concerning Möbius bands from unstretchable rectangular strips that Starostin and van der Heijden [6] set out to solve is of classical importance and they claimed that they solved it. Of course, such a claim would attract great attention, and coming from the continuum mechanics community and being interested in the theory of material surfaces, we were attracted. But, we found a major difficulty with their solution strategy that no one else had reported on — a difficulty serious enough to negate their claim. We have responded to their comments by illustrating a specific fundamental inconsistency in their work in a professional manner using standard principles of mechanics and elementary mathematical reasoning. There is no escaping the fact that the variational strategy that Starostin and van der Heijden [6] (amended with the comment of van der Heijden and Starostin [1]) applied in their approach to the classical Möbius band problem admits competing surface variations that stretch the material in contradiction to the hypothesis that the material be unstretchable. We have not stated in our reply that Starostin and van der Heijden [6] are wrong. Rather, we have responded to the specific steps that they have recorded in their comment [1] in order to support their position that we find to be mistaken. This involves noting a serious inconsistency which we leave to the reader to appraise.

Major point 3

Proposed resolution:

My feeling here is that the resolution is subtle. I don't believe it is a question of either of the papers [3] or [4] being "wrong". In my view [4] is a calculation that makes certain implicit assumptions, whereas [3] is a paper that would propose a more general theory to problems like this, which *may* lead to solutions that are more consistent with reality.

Chen and Fried claim in [1] that paper [3] was not intended to say that [4] (and other papers like it) are incorrect. Nevertheless van der Heijden and Starostin, the authors of [4], have taken exception to the word "erroneous" (and others like it) within [3]. This has led them to claim in [2] that Chen and Fried are wrong, which has led to an escalation.

My proposed resolution is a mild toning down of [2] to withdraw the claim that Chen and Fried are wrong and then for Chen and Fried to be invited to write a much more succinct version of their reply [1] that takes on board the fact that van der Heijden and Starostin are no longer claiming any errors in [1], but are simply pointing out that their theory is not as flawed as a reader of [1] might think.

While we appreciate the sentiments underlying the proposed resolution, our understanding is that the comment [1] has been accepted for publication in its present form. Nevertheless, we emphasize that we have done everything possible to present our case professionally and respectfully.

Response to Reviewer #3

In its current form, I do not deem the paper suitable for publication in Proceedings A. The reason is that the authors' critique of Starostin and van der Heijden's variational approach is far from being clear and neatly explained.

We have worked hard to clarify and neatly explain our perspective and findings in our revision of RSPA-2021-0856 to the comment [1] of van der Heijden and Starostin that is referenced therein.

The problem at issue is purely geometrical. It is clear that, in general, the standard parametrization of a developable surface does not represent the isometry mapping the surface to its planar development; equivalently, such parametrization is not an isometric embedding. However, one can use *any* parametrization to compute the bending energy (defined as the integral of the mean curvature squared).

We agree with the observations in the foregoing comment. In our reply below, [1], [5], and [6] refer to the corresponding references in our revision of RSPA-2021-0856.

Using the Wunderlich functional to compute the bending energy of a flat material surface that has been isometrically deformed is of course valid if that surface is known to admit a rectifying developable parametrization. However, an example provided in Section 5.1 of Chen, Fosdick and Fried [5] shows that the bending energy $E = 2\mu \int_{\mathcal{S}} H^2 da$ of a rectangular material strip \mathcal{D} that is isometrically deformed to a conical ribbon \mathcal{S} (see (5.1)–(5.11) of [5]) is underestimated by the Wunderlich functional E_W (see (5.15)–(5.16) of [5]) that results from restricting E to the class of rectifying developable parametrizations of \mathcal{D} unless the width-to-length ratio of \mathcal{D} is surprisingly small, in which case Sadowsky's dimensional reduction E_S (see (5.17)–(5.18) of [5]) of E provides a reasonable approximation to E . A construction in Section 5.3 of [5] shows that the bending energy E of an isometrically deformed rectangular material strip \mathcal{D} can be arbitrarily large but that the corresponding value of E_W always vanishes. In these examples, E_W underestimates E . Complementary examples for which E_W overestimates E can also be constructed.

For an isometric deformation of a rectangular material strip into an orientable or nonorientable band, the Wunderlich functional must be considered in conjunction with conditions that ensure that the rulings cover the band. Sufficient to ensure this is the requirement that the short edges $s = 0$ and $s = L$ of the rectangular reference configuration are material rulings. The analytical form of this requirement — namely that $\eta = \tau/\kappa$ satisfy $\eta(0) = 0$ and $\eta(L) = 0$ is not mentioned in the *Nature Materials* paper [6] of Starostin and van der Heijden. Instead, the discussion of the covering problem in [5] explains the need for conditions on η at $s = 0$ and $s = L$ to ensure that the planar development of a rectifying developable surface is a rectangle of appropriate dimensions. The conditions $\eta(0) = 0$ and $\eta(L) = 0$ were belatedly supplied in the comment [1] of van der Heijden and Starostin without acknowledging that they do not appear in their *Nature Materials* paper [6].

More precisely, suppose that \mathcal{S} is a developable Möbius strip and let \mathcal{D} be the planar development (i.e., isometric image) of \mathcal{S} . The problem is “merely” to find, among all developable Möbius strips having the same development \mathcal{D} , the one, say \mathcal{S}_* , that minimizes the bending energy. As the energy is invariant under reparametrization, the choice of parametrization employed in the process is completely irrelevant. Once we have \mathcal{S}_* , we know that there exists an isometry from \mathcal{S}_* to \mathcal{D} , and so we are done.

In the mechanics problem, the planar reference strip \mathcal{D} is not simply the development of a rectifying

developable Möbius band \mathcal{S} . It serves a higher purpose as the given reference configuration of a planar material strip wherein the points of the strip are identified and marked, once and for all, as material points. The fundamental problem is not purely geometrical. It is instead to find, among all Möbius bands \mathcal{S} that are isometric deformations from the same reference configuration of once and for all marked material points, the one developable Möbius band \mathcal{S}_* that minimizes the bending energy. Of course, if such a band \mathcal{S}_* is found, then it will have the property that its planar development has the shape \mathcal{D} . But importantly, in the minimization process that Starostin and van der Heijden adopted, they admitted only rectifying developable Möbius bands to compete for the minimum. All of their competitors have the same planar development \mathcal{D} but, in addition, they are supposed to preserve the distance between the identified and once and for all marked material points of \mathcal{D} and their images in \mathcal{S} . Therein lies the major issue with their variational strategy: their competing rectifying developable Möbius bands do not satisfy this distance preserving requirement.

It is therefore completely unclear to me how the authors are able to claim that Starostin and van der Heijden's variational approach does not give the correct solution. Note that, in their comment, Starostin and van der Heijden are not speaking about the variational scheme; they are only describing the isometry between a given \mathcal{S} and \mathcal{D} .

Because of this, the mechanics problem of finding a free-standing Möbius band made from isometrically deforming a single rectangular strip of an unstretchable two-dimensional material has not been solved. At best, they solve the differential geometry problem of finding the Möbius band that optimizes the Wunderlich functional among all rectifying developable Möbius bands that correspond to the planar development of a rectangle of length L and width $2w$. The mechanics problem contains more: It is concerned with an unstretchable material surface and the developable surfaces that compete in the optimizing variational scheme of Starostin and van der Heijden must respect that condition. The rectifying developable Möbius bands that they consider do not satisfy this constraint, and we comment on this in the second paragraph of Section 3.3 of our revision of RSPA-2021-0856.

As far as I understand, the desired fixed "reference configuration" is just the set \mathcal{D} , which has zero energy (in any coordinate system). The authors seem to have in mind two distinct problems: one "related to the differential geometry of rectifying developable surfaces"; the other "to the mechanics of unstretchable material surfaces". It would be nice if the authors could explain how these two problems are different.

The planar reference configuration of a material surface is the set \mathcal{D} of points, identified as material points, that are fixed and once and for all identified. If the surface is unstretchable, then the distance between each pair of material points cannot be changed. When \mathcal{D} is identified simply as the planar development (i.e., isometric image) of a set of developable Möbius strips then its interpretation as a unstretchable material surface is compromised because the planar development can remain fixed among the whole set of developable Möbius strips while internal distortions distinguish the various associated reference configurations. Please consider our response to the previous comments.

In conclusion, I suggest that the authors carefully reconsider their critique. In my opinion, to help the reader understand the authors' point of view, the paper should be less verbose and a bit more rigorous from a mathematical viewpoint.

The mathematical tools required for our arguments are elementary and we have made every effort to ensure that those arguments are as rigorous as needed to make our case.

Appendix C

Response to the reviews of RSPA-2021-0856.R1

Yi-chao Chen, Roger Fosdick and Eliot Fried

Response to Reviewer #1

Authors appear to have answered the issues adequately and I am happy that the paper is acceptable as is.

We appreciate the supportive statement from Reviewer #1.

Response to Reviewer #2

I thank the authors for toning down the rhetoric as I suggested. I have asked the editor if the authors of the original comment can also be asked to tone down their rhetoric too by making some further minor changes at the proof stage.

We once again thank the reviewer for encouraging us to tone down our rhetoric.

There is still one major point I disagree with. . . It is the sentence “Chen and Fried [2] do not state that the solution of Starostin and van der Heijden [6] is incorrect”. I disagree with the authors assertion that the authors state in their reply to my review that they did not state this even implicitly. Interpretation and meaning of phrases is something that is primarily at the discretion of the reader, not the author. I certainly got the impression that the word ”erroneous” when referring to formulations which include that in [6] meant that the authors of [2] were implicitly saying that the solution in [6] is “wrong”. I therefore ask the authors to rephrase this part of their response. . . something like

We appreciate that Starostin and van der Heijden have formed the impression that the authors of [2] were claiming that the solution of [6] is incorrect. We should like to point out though that this was not explicitly stated in [2], the intention of which was to explore whether. . .

We have incorporated the word “explicitly” in the first sentence of Section 2.

Response to Reviewer #3

In their response to the referee reports, the authors write that “at best, [Starostin and van der Heijden] solve the differential geometry problem of finding the Möbius band that optimizes the Wunderlich functional among all rectifying developable Möbius bands that correspond to the planar development of a rectangle of length L and width $2w$ ”. This is exactly what I have always thought was Starostin and van der Heijden’s contribution. (I agree that they could have added more information when explaining the method.)

We are pleased to be in agreement with the reviewer.

In my opinion, the authors’ statement is a rather curious admission: in “Möbius bands, unstretchable material sheets and developable surfaces”, Chen and Fried criticizes earlier work on developable surfaces without contemplating the possibility that some of it might not be addressing the authors’ intended mechanical problem, but rather the geometric one (which I suppose should deserve some respect too).

We concur that the problem of minimizing the Wunderlich functional over the class of rectifying developables is viable and of interest from a purely geometrical perspective. However, as we explain at the outset of Subsection 3.3, the goal of Starostin and van der Heijden was to consider a flat, undistorted rectangular material strip and to obtain from it the shape of a Möbius band formed by a material surface that can “deform [only] in such a way that its metrical properties are barely changed”. Their intention was to restrict their mechanical variational problem to unstretchable material surfaces, all of which originated from a given undistorted rectangular material strip. Thus, the admissible variations would need to be so constrained. The problem is that they restricted attention to rectifying developable Möbius bands noting that they all were developable into the same rectangular shape as was the given undistorted rectangular material strip. However, all but one are not isometric deformations of the given undistorted rectangular material strip, and so the admissible deformations in their mechanical variational problem are not all unstretchable.

As a matter of fact, the distinction between these two problems remains a complete mystery to me. The authors insist that, from the point of view of mechanics, Starostin and van der Heijden’s approach does not lead to the correct solution. They note that, albeit any two different surfaces competing in the variational scheme have the same planar development, it is not possible to map isometrically one to the other through their “rectifying parametrizations”. I fully agree with this observation, but I reject their conclusion, as the bending energy does not depend on the parametrization, only on its image. Hence, I do not see why the lack of a fixed reference configuration should be an issue.

The conclusion that the reviewer rejects is based on the understanding that the (dimensionless) bending energy

$$\int_S H^2 da \tag{1}$$

of S , which serves as the precursor of the Wunderlich functional, is to be minimized subject to the constraint that the deformation be unstretchable. This is an expression of the mechanical variational problem of Starostin and van der Heijden. To impose the material constraint, it is necessary to identify the positions of all material particles in some fixed reference configuration and to ensure that the distance between any pair of material particles not change under any deformation. Even

though the value of (1) does not depend on the parametrization of \mathcal{S} , it remains necessary to respect the constraint in the analysis of the variational problem and to do so it is essential to be given a fixed reference configuration as described above. As the reviewer agrees, rectifying developables allow for stretching. Hence, minimizing (1) over that admissible class of deformations does not ensure that the constraint is met.

In their comment, van der Heijden and Starostin claim that the $[0, L] \times [-w, w]$ rectangular region they introduce serves as a reference configuration and perform a calculation to show that a band of length L and width $2w$ with a specific rectifying developable parametrization can be used to parametrize an isometric deformation of the $[0, L] \times [-w, w]$ rectangular region into the band. In so doing, they appear to accept the importance of introducing a given fixed reference configuration as described in the preceding paragraph. (Despite this point of agreement, we emphasize that, as recognized by the reviewer, the planar developments of any two bands of length L and width $2w$ with distinct rectifying developable parametrizations onto the same rectangular shape $[0, L] \times [-w, w]$ need not imply that there is an isometric deformation of one of the parametrizations into the other.) The reliance upon a given fixed reference configuration is consistent with the understanding, expressed above, that their goal was to solve the “mechanical problem” rather than the “geometric problem”.